# Loss of Smell and Taste Can Accurately Predict COVID-19 Infection: A Machine-Learning Approach

**DOI:** 10.3390/jcm10040570

**Published:** 2021-02-03

**Authors:** María A Callejon-Leblic, Ramon Moreno-Luna, Alfonso Del Cuvillo, Isabel M Reyes-Tejero, Miguel A Garcia-Villaran, Marta Santos-Peña, Juan M Maza-Solano, Daniel I Martín-Jimenez, Jose M Palacios-Garcia, Carlos Fernandez-Velez, Jaime Gonzalez-Garcia, Juan M Sanchez-Calvo, Juan Solanellas-Soler, Serafin Sanchez-Gomez

**Affiliations:** 1Rhinology Unit, Department of Otolaryngology, Head and Neck Surgery, Virgen Macarena University Hospital, 41009 Seville, Spain; amparocallejon@gmail.com (M.A.C.-L.); ramoluorl@gmail.com (R.M.-L.); juan.maza.solano@gmail.com (J.M.M.-S.); danimartin12@live.com (D.I.M.-J.); josepalaciosgarcia@hotmail.com (J.M.P.-G.); carlosfervel@gmail.com (C.F.-V.); xaimeglez@gmail.com (J.G.-G.); serafin.sanchez.sspa@gmail.com (S.S.-G.); 2Biomedical Engineering Group, University of Seville, 41092 Seville, Spain; 3Rhinology and Asthma Unit, ENT Department, The University Hospital of Jerez, 11407 Jerez de la Frontera, Cadiz, Spain; 4Rhinology Unit, Department of Otolaryngology, Virgen de Valme University Hospital, 41014 Seville, Spain; isareyes6@hotmail.com (I.M.R.-T.); magarciavillaran@gmail.com (M.A.G.-V.); juan.solanellas.sspa@juntadeandalucia.es (J.S.-S.); 5COVID-19 Unit, Infectious Disease Department, The University Hospital of Jerez, 11407 Jerez de la Frontera, Cadiz, Spain; martasantospe@gmail.com (M.S.-P.); jmanuel.sanchez.sspa@juntadeandalucia.es (J.M.S.-C.)

**Keywords:** COVID-19, machine learning, prediction model, SARS-CoV-2, smell, taste, visual analog scale

## Abstract

The COVID-19 outbreak has spread extensively around the world. Loss of smell and taste have emerged as main predictors for COVID-19. The objective of our study is to develop a comprehensive machine learning (ML) modelling framework to assess the predictive value of smell and taste disorders, along with other symptoms, in COVID-19 infection. A multicenter case-control study was performed, in which suspected cases for COVID-19, who were tested by real-time reverse-transcription polymerase chain reaction (RT-PCR), informed about the presence and severity of their symptoms using visual analog scales (VAS). ML algorithms were applied to the collected data to predict a COVID-19 diagnosis using a 50-fold cross-validation scheme by randomly splitting the patients in training (75%) and testing datasets (25%). A total of 777 patients were included. Loss of smell and taste were found to be the symptoms with higher odds ratios of 6.21 and 2.42 for COVID-19 positivity. The ML algorithms applied reached an average accuracy of 80%, a sensitivity of 82%, and a specificity of 78% when using VAS to predict a COVID-19 diagnosis. This study concludes that smell and taste disorders are accurate predictors, with ML algorithms constituting helpful tools for COVID-19 diagnostic prediction.

## 1. Introduction

The COVID-19 outbreak has been the most threatening challenge healthcare systems have faced in modern times, with the novel coronavirus SARS-CoV-2 spreading rapidly around the world, causing more than 2 million deaths in a year according to the World Health Organization (WHO), as well as an incalculable socioeconomic burden.

Diagnostic testing for COVID-19 has been shown to be critical to track the spread of the virus, understand the epidemiology, inform case management, and eventually reduce transmission. In order to clarify priorities for testing, the WHO has provided definitions for suspected, probable, and confirmed cases, as well as for contacts of the cases, which were subsequently adopted by Centers for Disease Control and Prevention (CDC). Furthermore, in order to optimize testing procedures, especially in areas with limited healthcare resources, the identification of the most predictive symptoms would help select those individuals with higher probabilities of being infected, establish guidelines for self-isolation, and eventually contribute to the suppression of the spread of the virus. 

Although SARS-Cov-2 accesses the respiratory system with most people experiencing a mild or subclinical disease, especially in the early stages, clinical manifestations of COVID-19 are broad and include common symptoms such as malaise, fever, cough, shortness of breath, myalgia, sore throat, headache, nausea, or diarrhea [1,2]. Nevertheless, loss of smell and taste have emerged as the most prevalent and predictive symptoms in mild COVID-19 cases [3,4]. Smell and taste disorders that are related to upper respiratory tract infections are caused by more than 200 viruses, of which, 10-15% are coronaviruses [5]. There is a need for studies that clarify the actual prevalence of smell and taste disorders in COVID-19, which vary from 5.1% to 87% in different studies [5,6,7,8,9,10,11,12,13], possibly due to ethnical and geographical differences [3,11]. Moreover, the symptoms of COVID-19 infection may overlap with those of other common diseases such as allergies or routine viral illnesses, making it difficult to distinguish between them.

Prediction models based on symptoms such as loss of smell and taste have been proposed as helpful tools to predict COVID-19 diagnosis [9,10,14,15] as well as early indicators of the effectiveness of containment measures in new outbreaks [16]. New approaches based on machine learning (ML) techniques have also attracted attention from researchers, who have recently used them to analyze and predict olfactory dysfunction in other nasal pathologies [17]. ML methods draw on algorithms that can be applied to a population-based systems approach, modelling complex interactions and associations between multiple variables, showing a higher prediction accuracy when compared with classical statistical methods. Therefore, this study focuses on two main objectives: (1) develop a comprehensive machine learning (ML) modelling framework to assess the predictive value of smell and taste disorders in the COVID-19 disease; and (2) analyze the prevalence and association of loss of smell and taste, in combination with other common symptoms, with a COVID-19 diagnosis.

## 2. Materials and Methods

### 2.1. Study Design, Setting, and Participants

This was a multicenter, cross-sectional case-control study of subjects aged 18 or older that were suspected of having COVID-19, i.e., symptomatic or asymptomatic subjects in close contact with a positive case, according to the recommendations from the WHO. The subjects underwent laboratory testing via RT-PCR by collecting upper respiratory tract specimens (nasopharyngeal and oropharyngeal). All subjects filled out a questionnaire regarding the presence and the severity of their symptoms, including loss of smell and taste. Those who were RT-PCR positive were assigned to the case group and those who tested negative to the control group. The data was collected in four different hospitals in the Spanish regions of Seville and Cadiz, between March and April 2020. This study complies with the Declaration of Helsinki and was approved by the Ethics Committee of the Virgen Macarena University Hospital in Seville, Spain. All participants were informed about the study and gave their consent to participate. 

### 2.2. Study Variables

In addition to demographic variables such as age and sex, the presence of symptoms such as loss of smell, loss of taste, nasal obstruction, nasal discharge, facial pain, cough, and dyspnea were reported through categorical variables, and their severity was reported through validated visual analog scales (VAS) ranging from 0 to 100. Fever and diarrhea were also informed as categorical variables.

### 2.3. Data Analysis and Statistical Methods

Statistical analysis was performed using the Statistics and Machine Learning Toolbox in Matlab (v. R2018a, The MathWorks Inc, Natick, MA, USA). A single mean and mode imputation method, adding randomly generated values from a normal distribution with mean 0 and deviation 7.5, were used to impute continuous and categorical missing variables, respectively [18]. We first compared the prevalence of categorical symptoms between positive and negative COVID-19 groups using the χ^2^ test. To also analyze the differences in the intensity of the symptoms reported between the two groups, Student’s t-test was used with numerical VAS variables. In our analysis, a *p*-value of 0.05 or lower was considered significant. For each symptom, the VAS cut-off value that better predicted COVID-19 positivity was calculated by maximizing the Youden index using receiver operating characteristic (ROC) curves. We then recoded the continuous VAS variables into new dichotomous ones, indicating whether the VAS score was higher or lower than the corresponding cut-off value. Subsequently, the association of each symptom with a COVID-19 diagnosis was assessed through bivariate regression analysis. 

To identify the symptoms most strongly associated with a COVID-19 diagnosis, we performed backward and forward step-wise multivariate logistic regression (LR) algorithms using both Akaike (AIC) and Bayesian (BIC) information criteria with the dichotomized VAS dataset. We performed a holdout validation by randomly splitting the sample into training and testing datasets in a ratio of 75:25. The performance of the obtained model was assessed in terms of parameters such as the area under the curve (AUC), sensitivity (SE), specificity (SP), positive predictive value (PPV), and negative predictive value (NPV). For the sake of clarity, the details of the model implementation and the holdout validation can be seen in Figure 1a.

### 2.4. Machine Leaning (ML) Approach to Predict a COVID-19 Diagnosis

To assess the predictive value of smell and taste disorders in the COVID-19 disease, we develop a comprehensive machine learning (ML) modelling framework including logistic regression (LR) and two other ML algorithms, Random Forest (RF) and Support Vector Machine (SVM), which were respectively implemented in order to evaluate their discrimination accuracy when predicting a positive COVID-19 test result. For RF algorithm, a bagged ensemble of 200 regression trees was trained to estimate predictor score values using a curvature test, while a basis kernel approach was chosen for SVM. Five different datasets including categorical variables, continuous VAS, and dichotomized VAS were considered with the aim of assessing the discrimination ability of different types of variables when predicting COVID-19 infection. Specifically, (1) a first dataset considering the (categorical) presence of the nine symptoms queried in the study (Dataset 1); (2) a second dataset containing the (continuous) VAS scores reporting the intensity of the symptoms (Dataset 2); and (3) a third dataset with dichotomized VAS variables reporting whether the intensity of symptoms was higher or lower than a corresponding VAS cut-off value (Dataset 3). Datasets 1–3 also included sex and age as predictor variables. Afterwards, we also analyzed the accuracy of simplified models containing a lower number of predictors, such as (4) a reduced model considering four symptoms (loss of smell, loss of taste, fever, and diarrhea) (Dataset 4); and (5) a parsimonious model including only two symptoms (loss of smell and fever) (Dataset 5). A cross-validation scheme was designed by randomly splitting the sample in training (75%) and testing (25%) datasets 50 times (i.e., 50-fold validation) (See Figure 1b). The discrimination accuracy of the different datasets and ML algorithms tested was evaluated in terms of mean AUC, SE, SP, PPV, and NPV parameters over 50 training/test split combinations. Therefore, we obtained 50 different train/test splits for each of the five predictor datasets analyzed and each of the three ML algorithms considered (LR, RF and SVM), which yields to a total number of 750 (5 × 3 × 50) model datasets computed with this approach. Finally, permutation-based importance score ranking was used to identify the most important predictor symptoms in RF algorithm.

## 3. Results

### 3.1. Demographic Characteristics of the Sample

The study included data from 777 subjects suspected of suffering from COVID-19, collected in four different hospitals in the Spanish regions of Seville and Cadiz between March and April 2020. Of these, 421 cases were positive via RT-PCR, and 257 (61%) were female. Out of the 356 controls, 277 (78%) were female. There were no relevant differences in age between cases and controls, and more women than men participated in the study (69 vs. 31%). There were 64 patients out of 777 who had one or more missing values. Overall, 103 missing values out of 6993 were imputed (1.5%).

### 3.2. Prevalence and Intensity of Symptoms between Cases and Controls

Table 1 shows the prevalence of symptoms for the 777 subjects included in the study. There were 102 asymptomatic subjects, of which only 15 were positive for COVID-19. The most common symptoms in subjects who tested positive for COVID-19 were cough (73%), loss of taste (63%), fever (61%), and loss of smell (61%). Significant differences (*p* < 0.001) between cases and controls were found in the prevalence of all symptoms, except for nasal obstruction and discharge. Figure 2 shows the distribution of continuous variables (age and self-reported VAS scores for symptoms) in each of the two groups. Interestingly, positive COVID-19 subjects reported a more intense loss of smell (51.4 ± 44.9 versus 7.9 ± 21.8; *p* < 0.0001) and taste (49.6 ± 41.9 versus 8.7 ± 21; *p* < 0.0001) than negative subjects. In addition, 278 out of the 312 subjects who reported loss of smell also reported loss of taste (236 positive), and 85 reported both losses with a maximum intensity of 100 (82 positive). A high correlation was found between loss of smell and loss of taste, with a Pearson coefficient of 0.83 in the complete sample and of 0.79 in the positive COVID-19 group. However, no correlation was found between loss of smell and any other symptoms, not even those related to the nasal pathway, such as nasal obstruction and discharge. Significant differences in the intensity reported for facial pain, cough, and dyspnea between groups were also seen (*p* < 0.0001). Although still significant, the difference in the intensity reported between groups for nasal obstruction was considerably lower (20.7 ± 31.7 versus 16 ± 27.9; *p* = 0.0311). No differences were found for nasal discharge between groups (25.1 ± 30.7 versus 22 ± 28.5; *p* = 0.1481). 

### 3.3. VAS Cut-off Points That Optimally Predicted COVID-19 Diagnosis

The VAS cut-off points that optimally predicted COVID-19 diagnosis are shown in Table 2, together with the crude ORs and confidence intervals derived from the bivariate regression analysis. Table 2 also shows the discrimination accuracy, measured as the area under the curve (AUC), together with the sensitivity (SE) and specificity (SP) calculated for each symptom. Loss of taste (OR = 12.62; CI 95%: 8.50–18.73; *p* < 0.0001) and loss of smell (OR = 10.85; CI 95%: 7.47–15.77; *p* < 0.0001) were the symptoms showing the highest ORs and discrimination accuracy (AUC = 0.76). Cough and dyspnea showed an AUC equal to 0.65 and 0.61, respectively, following loss of taste and smell. Nasal obstruction (AUC = 0.53), nasal discharge (AUC = 0.52) and facial pain (AUC = 0.57) were not predictive of COVID-19 in the regression analysis.

### 3.4. Multivariate Logistic Regression (LR) Model

A multivariate COVID-19 prediction model was derived by performing logistic step-wise backward-forward regression considering the nine dichotomized symptom variables as well as sex and age as predictors. The model with the lowest Bayesian Information Criterion (BIC), see equation below, was selected as the best model:x = −1.76 + 0.88 × ((1 if VAS for loss of smell ≥ 21) or (0 if VAS for loss of smell < 21)) + 1.83 × ((1 if VAS for loss of taste ≥ 44) or (0 if VAS for loss of taste < 44)) + 0.79 × ((1 if VAS for dyspnea ≥ 28) or (0 if VAS for dyspnea < 28)) + 0.61 × ((1 if fever) or (0 if no fever)) + 0.70 × ((1 if diarrhea) or (0 if no diarrhea)) − 1.13 × ((1 if female) or (0 if male))(1)

In this model, loss of taste, loss of smell, fever, diarrhea, and dyspnea were positively and independently associated with a positive COVID-19 test, while being female was negatively associated. Other variables and symptoms such as age, nasal obstruction, nasal discharge, facial pain, and cough were not associated with a COVID-19 diagnosis. Odds ratios, confidence intervals, and *p*-values are shown in Table 3. Alternatively, a forest plot diagram can be seen in Figure 3a. Loss of taste and smell were the two symptoms that exhibited the highest ORs. Specifically, subjects reporting a VAS score greater than 21 for loss of smell had an odds ratio 2.7 times higher of being positive for COVID-19 (OR = 2.7; CI 95%: 1.42–5.12; *p* = 0.002). This odds ratio increased up to 6 times for those that reported a VAS score greater than 44 for loss of taste (OR = 6.02; CI 95%: 3.10–11.69; *p* < 0.001). The x value obtained from this LR model (Equation 1) was converted into probabilities through the exp(x)/(1 + exp(x)) transformation. In our analysis, we assumed that a positive COVID-19 case is predicted when the probability is greater than 0.5. Figure 3b shows ROC curves for COVID-19 prediction when the BIC-derived LR model was assessed with the 25% testing dataset. In the testing set, the prediction model had an area under the curve (AUC) of 0.78 (0.72–0.83), a sensitivity of 0.72 (0.69–0.75), a specificity of 0.84 (0.82–0.87), a positive predictive value of 0.83 (0.80–0.85), and a negative predictive value of 0.74 (0.71–0.77). When selecting a model with the lowest Akaike criterion in the stepwise regression analysis, similar values of classification performance were obtained (AUC = 0.78, SE = 0.76, SP = 0.80, PPV = 0.80, NPV = 0.76). However, the AIC model also included nasal obstruction with an OR of 0.63, indicating that subjects that reported a more severe nasal obstruction (with a VAS score higher than 52) were more likely to test negative for COVID-19.

### 3.5. ML Results: Comparison of Accuracy between Algorithms and Model Datasets

Table 4 lists mean AUC, SE, SP, PPV, and NPV values obtained for each dataset and ML algorithm considered. Alternatively, Figure 4 shows boxplots for the distribution of the 50-fold cross-validation estimates for AUC, SE, and SP values; for the various datasets (Datasets 1–5) and ML algorithms (LR, RF, and SVM) considered. Dataset 2 (continuous VAS scores for the intensity of symptoms) was the most predictive set of variables for COVID-19, with mean AUC values near 0.78 for LR and 0.80 for RF and SVM algorithms, respectively. Dataset 3 (dichotomized VAS) was the next most predictive dataset with similar mean AUC values (0.77 for LR and 0.79 for RF and SVM). A predictor dataset that reported only the presence or absence of the symptom (Dataset 1) obtained lower mean AUC values between 0.76 and 0.78, which also showed to be the least specific, with mean SP values of 0.72 for LR and SVM and of 0.75 for RF algorithms. Although Datasets 4 and 5, which included only four and two symptoms respectively, had lower mean AUC values than the other datasets, especially for RF and SVM, they were still higher than 0.75. Finally, permutation-based importance scores calculated for RF algorithm ranked loss of taste, loss of smell, fever, and diarrhea as most relevant symptoms for COVID-19 diagnosis.

## 4. Discussion

We have shown in our study the high accuracy of smell and taste disorders in predicting the COVID-19 disease using different predictive analysis models. The main outcomes from our study are (1) loss of smell and taste are highly prevalent symptoms and are strongly associated with COVID-19 positivity. (2) VAS reporting the intensity of loss of smell and taste were the most relevant predictors for COVID-19 diagnosis. (3) ML algorithms can accurately predict COVID-19 diagnosis, thus constituting helpful tools to identify subjects with a higher probability of being infected.

### 4.1. Prevalence of Smell and Taste Disorders in COVID-19 Subjects

Many authors have highlighted the high prevalence of loss of smell and taste in COVID-19 patients. A meta-analysis demonstrated a prevalence of 52.73% (95% CI: 29.64–75.23%) and 43.93% (95% CI: 20.46–68.95%) for olfactory and gustatory dysfunction among COVID-19 patients, reaching 86.60% in studies that used validated instruments [19]. Another meta-analysis, including 42 studies and 38,198 patients, showed an estimated random prevalence of olfactory dysfunction of 43%, taste dysfunction of 44.6%, and overall chemosensory dysfunction of 47.4%. Both the ethnic groups and the geographical regions where the virus has spread have shown to play a fundamental role in the variability observed for the prevalence of chemosensory disorders, reported to be 3–6 times higher in Caucasians than in East Asians [11]. In our study, based on the south of Spain Caucasian population, a prevalence of 61% for loss of smell in positive COVID-19 patients versus that of 15% in controls was found. For loss of taste, a prevalence of 63% versus 19% was found, thus showing highly significant differences between groups. These results corroborate those found in other similar case-control studies, such as one that included 1480 patients suffering from influenza-like symptoms, where loss of smell and taste were reported in 68% and 71% of COVID-19 positive subjects, respectively, compared to 16% and 17% of COVID-19-negative patients, respectively (*p* < 0.001) [20]. Similarly, in a multicenter cross-sectional study including 989 patients, smell and taste dysfunction were at least twice as common in COVID-19 positive cases than in controls, with more than a half suffering from a severe loss of smell or taste, and more than 90% suffering from both impairments [21]. We observed a similar percentage of subjects (89%) that reported both loss of smell and taste, 30% of which reported the maximum VAS score of 100 for both disorders. A systematic review and meta-analysis reported an overall prevalence of smell or taste dysfunction of 47%, with an estimate of 67% in mild-to-moderate symptomatic patients, similar to that found in our study. This study also concluded that loss of smell and taste appeared earlier than other symptoms in 20% of cases and concomitant in 28% [22]. Systematic reviews have shown frequencies ranging between 22 and 86% for anosmia and 33 and 56% for dysgeusia [23,24]. Although women have been reported to be more often affected by these disorders, the high variability observed in studies has not allowed authors to identify a clear relationship between the prevalence of these disorders and gender [11]. Our results showed a prevalence of 65% and 55% for loss of smell in women and men, respectively, increasing to 71% and 51%, respectively, for loss of taste. 

### 4.2. Predictive Value of Smell and Taste Disorders in the COVID-19 Disease

Loss of smell and taste have been shown to be strongly associated with COVID-19. In a study with 2,618,862 participants who reported their symptoms on a smartphone-based app, those who reported loss of smell and taste had an odds ratio of 6.74 (95% CI: 6.31–7.21) of being positive for COVID-19 [9]. Similarly, a systematic review including 2757 patients showed those that reported loss of smell and taste had 6-times higher odds of being positive for COVID-19, and those suffering from anosmia and ageusia had an odds ratio 10-times higher [23]. In line with these results, we have shown in our study a crude odds ratio of 10.9 (95% CI: 7.5–15.8) and 12.7 (95% CI: 8.50–18.7) for loss of smell and taste, respectively, of being positive for COVID-19 when compared with a negative control group.

Despite their high or low prevalence in patients, common COVID-19 symptoms have also been studied in order to assess their independent association with a positive diagnosis. In a multivariate analysis of 302 cases and controls, smell or taste changes, fever, and body ache were associated with COVID-19 positivity, whereas shortness of breath and sore throat were associated with a negative test result (*p* < 0.05) [10]. In [9], the authors performed a multivariate analysis concluding that loss of smell and taste, fatigue, persistent cough, and loss of appetite was the combination of symptoms most strongly correlated with COVID-19. Other symptoms also included fever, diarrhea, shortness of breath, delirium, abdominal pain, chest pain, and hoarse voice. In our study, the symptoms most strongly associated with COVID-19 positivity were loss of taste and smell, fever, diarrhea and dyspnea (see Figure 3). On the other hand, age, nasal obstruction, nasal discharge, facial pain, and cough were not independently associated with COVID-19. Although the prevalence of loss and taste disorders was greater in women than in men in our study, being a woman was found to be negatively associated with COVID-19 positivity (OR = 0.32; 95% CI: 0.20–0.51), in line with previous studies where women were found to be less likely to test positive [9,25]. In addition, based on AIC criterion, our LR model showed an OR of 0.63 for nasal obstruction (OR: 0.63, 95% CI: 0.34–1.15), which agrees with previous systematic reviews that found that pharyngodynia, nasal congestion, and rhinorrhea were less frequent symptoms in COVID-19 patients [26]. Although underlying mechanisms require further elucidation, the higher association of loss of smell compared with other common sinonasal symptoms, such as rhinorrhea or nasal obstruction, may suggest a greater affectation of the olfactory neuroepithelium due to COVID-19 [27,28,29]. The comparatively enhanced human airway expression of ACE2 (the binding mechanism that mediates SARS-CoV-2 entry into cells) in the olfactory neuroepithelium has been suggested as a possible cause of the neurological symptoms observed in COVID-19 patients and a main point of access for the virus into the central nervous system (CNS) [30]. The involvement of facial, glossopharyngeal, and the vagus nerve in the oral mucosa, which is responsible for the transport of taste signals into the solitary nucleus in the brainstem, has also been proposed to be the reason behind taste dysfunction in up to 88% of the COVID-19 patients [8]. The high neurological susceptibility to COVID-19 can be explained by the fact that the virus uses various potential routes to access the brain, including the respiratory and gastrointestinal tracts [31]. In addition, SARS-Cov-2 has been found in tears and conjunctival secretions [32], with ocular manifestations also suggesting the eyes to be an entry for the virus through the trigeminal nerve into the CNS [33]. However, other inflammatory mechanisms have been proposed for the dysgeusia, both in the presence or absence of olfactory symptoms, as an early or sole presentation of COVID-19 before the lungs or other organs are infected [34]. In our study, 278 out of the 312 subjects who reported loss of smell also reported loss of taste, with a correlation of 0.79 among positive COVID-19 patients.

### 4.3. Prediction Models for COVID-19 Diagnosis Based on Smell and Taste Disorders 

Prediction models that combine several variables or features to estimate the risk of being infected with COVID-19 have been proposed as useful tools to assist medical staff in triaging patients, especially when allocating limited healthcare resources. In Table 5, we summarize the main prediction models in the literature that have analyzed the predictive value of loss of smell and taste [9,10,14,15,35], and we compare their results with those obtained in our study. The studies reported in the literature are mainly based on stepwise LR models and have reported accuracy values ranging from 0.63 to 0.82. Details on the sample size, demographic characteristics, testing, and validation methods, as well as predictor variables, are comparatively listed in Table 5. In our study, we have also developed an LR model which included loss of smell, loss of taste, dyspnea, fever, diarrhea, and gender as significant predictor variables for COVID-19, reaching a classification accuracy (AUC) of 0.78, a sensitivity (SE) of 0.72, and a specificity (SP) of 0.84. We have furthered these results by implementing a COVID-19 prediction workflow including other ML algorithms such as RF and SVM to test the performance of different datasets as well as the importance of different predictor variables. Unlike previous works that reported prediction models mainly based on LR algorithms and the presence or absence of symptoms [9,10,14], the ML workflow designed in this study detected higher discrimination accuracy for continuous VAS predictors quantifying the intensity of symptoms (See Table 4). This agrees with preliminary results in [15], where VAS reached greater AUC values than categorical questions when predicting COVID-19. As expected, different datasets including different predictor variables and/or a reduced number of symptoms led to different accuracy values (see Figure 4), with each model reaching a different trade-off between AUC, SE, SP, PPV, and NPV values. An ML framework evaluating the accuracy of different datasets might help to define better screening strategies in different outbreak scenarios and populations. For instance, a parsimonious model that considered only the intensity of the loss of smell and the presence of fever reached a mean accuracy of 0.77 and a greater sensitivity of 0.81 in our study, and it still showed a high specificity of 0.72. More importantly, for all datasets assessed, our ML analysis approach confirmed loss of smell and taste as the two main predictors for COVID-19.

### 4.4. Limitations

A recent systematic review concluded that most prediction models for COVID-19 are poorly reported, at high risk of bias, and probably report an over-optimistic performance [36,37,38]. Therefore, there are many limitations when transferring the prediction models into the clinical practice, which cast doubt on their true applicability, especially for different outbreak scenarios and/or populations. In the context of a global pandemic where millions of patients have been infected, our study includes a relatively low sample size of 777 patients who were recruited while the COVID-19 outbreak spread rapidly. This has probably led to a higher positive predictive value in our models. In this sense, we have tried to improve the quality and applicability of the methods reported in this study by designing a comprehensive ML cross-validation workflow, calculating the mean value and the deviation of classification parameters (AUC, SE, SP, PPV, and NPV) over 50 cross-validated training and testing cohorts. Furthermore, we have implemented different ML algorithms (LR, RF, and SVM) and evaluated the performance of different datasets including categorical, continuous, and dichotomized VAS predictors.

Prediction models are aimed at screening potential COVID-19 cases that should be tested. Therefore, the misclassified cases from the proposed rules could potentially expose others. In this study, we have reported the cut-off points in the ROC curve that reached a trade-off between sensitivity and specificity by maximizing the Youden Index. An alternative approach aimed at maximizing the sensitivity and reducing the number of misclassified positive cases may be achieved by lowering the cut-offs here reported, albeit at a cost of reducing the specificity and accuracy of the prediction rules proposed. A major limitation of our study is that we have not included asymptomatic cases randomly selected from the general population, which would probably have changed the performance and predictive values of our models. The asymptomatic cases included were those considered suspected cases according to the criteria given by the WHO. In addition, there is a high disparity reported in the performance of COVID-19 RT-PCR diagnostic testing [39]. We only considered the first test performed to classify a subject as a positive case or control, which probably diminished the likelihood of being COVID-19 positive. It is recognized that a single negative RT-PCR test may sometimes be insufficient to confirm negativity.

Lastly, we consider that the quantitative assessment of symptoms by means of VAS allowed us to increase the discrimination accuracy of the prediction models performed, as demonstrated with the ML workflow proposed in this study. In addition, we were also able to find the cut-off point values that better predicted COVID-19 positivity, which allowed us to obtain meaningful conclusions regarding the association between the intensity of symptoms, such as loss of smell and taste, and COVID-19 diagnosis. However, we also recognize that VAS, although validated, cannot replace objective measures for the assessment of taste or smell sensation. 

## 5. Conclusions

Our study concludes that loss of smell and taste can accurately predict the COVID-19 disease. This work reviews and furthers the use of prediction models and machine learning algorithms for COVID-19 diagnosis, and the high predictive value of smell and taste disorders in the disease. These models may constitute helpful tools to optimize the screening of suspected cases in the context of new COVID-19 outbreaks. Through this study, we have demonstrated that a robust machine learning workflow can be used to accurately predict a COVID-19 diagnosis through a combination of symptoms and features, from which loss of smell and taste were the most relevant.

## Figures and Tables

**Figure 1 jcm-10-00570-f001:**
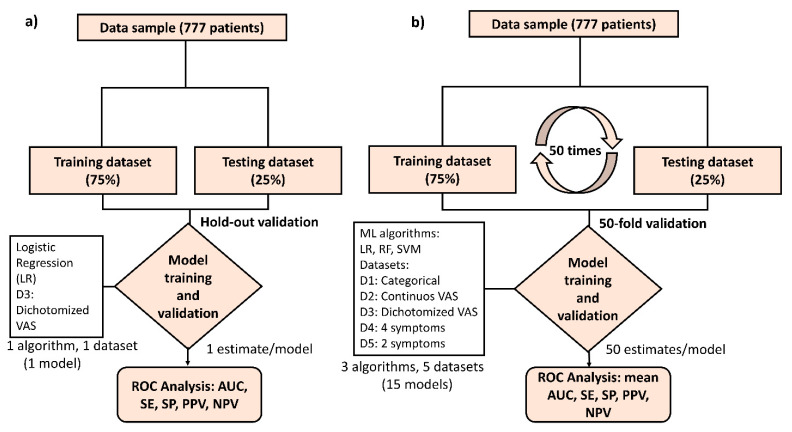
Modelling framework for the analysis of symptom associations and COVID-19 prediction. Data from 777 patients were obtained from different hospitals in the South of Spain. (**a**) For the analysis of the association between the intensity reported for loss of smell and taste, along with other symptoms, and a COVID-19 diagnosis, a first model was derived using step-wise logistic regression (LR) with a holdout validation scheme, by splitting the sample into a training (75%) and a testing dataset (25%). The performance of the model was assessed through ROC analysis, with AUC, SE, PPV and NPV parameters being calculated for the holdout testing (25%) dataset. (**b**) For the analysis of the discrimination ability and predictive value of different symptom variable datasets, including categorical (D1), continuous visual analog scales VAS (D2), dichotomized VAS (D3) as well as simplified predictor datasets with a reduced number of symptoms (D4 and D5), a comprehensive 50-fold cross-validation scheme was designed by assessing three different ML algorithms (LR, RF, and SVM). The performance of the models obtained were calculated through the mean AUC, SE, SP, PPV and NPV values over the 50-cross validated estimates obtained for each model tested. LR = logistic regression. RF = random forest. SVM = support vector machine, ROC= receiver operating characteristic, AUC= area under the curve, SE = sensitivity, SP = specificity, PPV= positive predictive value, NPV= negative predictive value.

**Figure 2 jcm-10-00570-f002:**
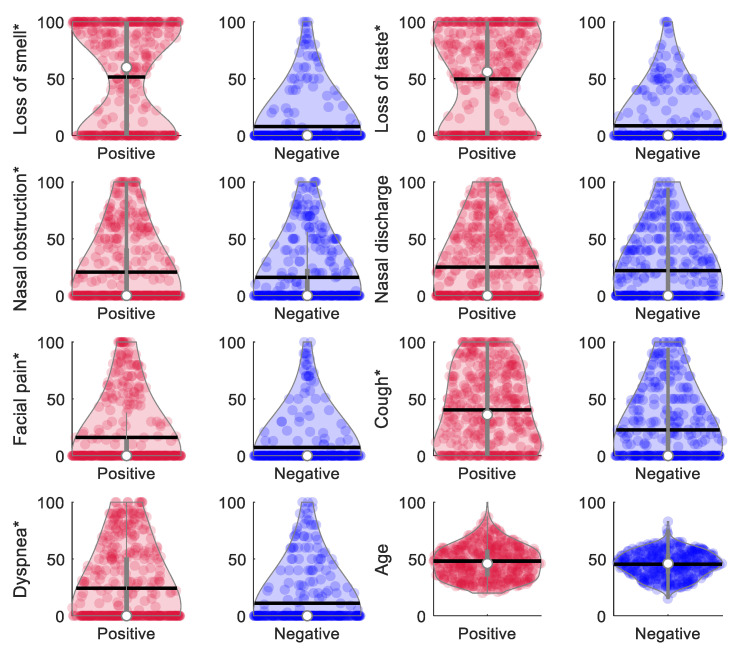
Violin plots showing the distribution of self-reported VAS scores for symptom intensity. The white dots depict the median value and the vertical gray lines the interquartile range (25th and 75th quantiles). Horizontal black lines represent the mean value. Symptoms accompanied with an asterisk (*) showed significant differences in the Student’s *t*-test between positive and negative groups: loss of smell, loss of taste, facial pain, cough, dyspnea (*p* < 0.0001), and nasal obstruction (*p* = 0.0311). Nasal discharge (*p* = 0.1481) and age (*p* = 0.0628) were not significantly different between groups.

**Figure 3 jcm-10-00570-f003:**
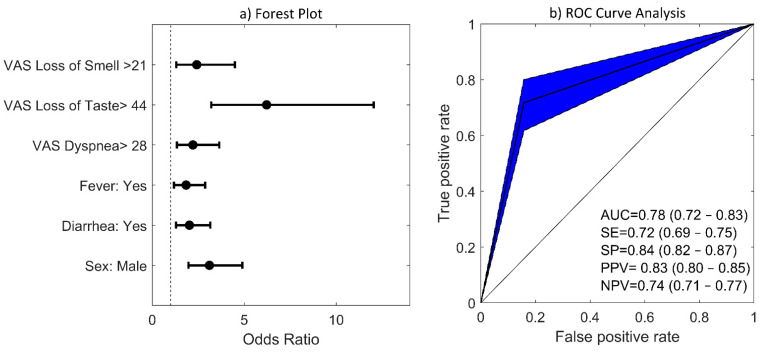
Step-wise logistic regression model obtained under BIC criterion: (**a**) Forest plots for association between symptoms and COVID-19 diagnosis. Note that for this analysis, VAS numeric variables reporting the intensity of symptoms were dichotomized into two categories: being higher or lower than the corresponding VAS cutoff point. VAS cutoff points for each symptom were previously calculated using ROC analysis and are listed in Table 2. Error bars denote 95% CIs; (**b**) ROC curve for prediction of COVID-19 in the (25%) holdout testing dataset. AUC, SE, SP, PPV, and NPV mean values are shown together with their 95% CIs. The confidence region of the ROC curve is depicted in blue.

**Figure 4 jcm-10-00570-f004:**
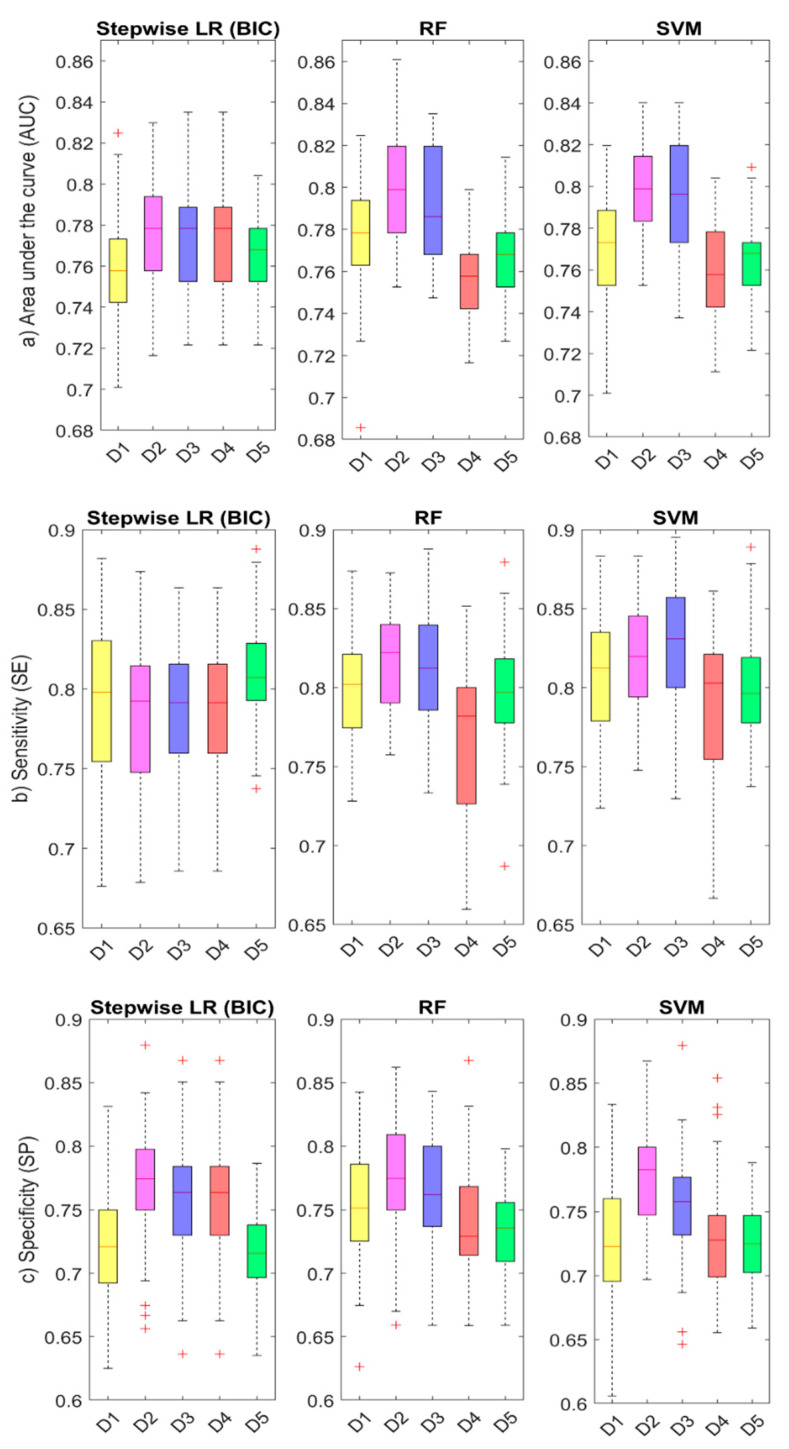
Comparison of accuracy for different predictor datasets and ML algorithms: the boxplots show the distribution of (**a**) AUC, (**b**) SE, and (**c**) SP parameters for different predictor datasets (D1–D5, see description in Table 4) and ML algorithms (LR, RF, and SVM). The red line represents the mean value (calculated over the 50-fold cross-validation estimates), and the edge boxes the 25th and 75th percentiles. The whiskers represent the minimum and maximum data values not considered outliers, and the outliers are plotted individually using the ‘+’ symbol in red.

**Table 1 jcm-10-00570-t001:** Prevalence of symptoms in the sample. Odds ratios and 95% CIs values were calculated using the counts (%) in the table. *p*-values were calculated via χ² test.

Symptom	Total (*n* = 777)	Cases (*n* = 421)	Controls (*n* = 356)	Odds Ratio	CI 95%	*p*-Value
Loss of smell	312 (40%)	257 (61%)	55 (15%)	8.58	[6.06, 12.14]	<0.0001
Loss of taste	332 (43%)	266 (63%)	66 (19%)	7.54	[5.41, 10.52]	<0.0001
Nasal obstruction	262 (34%)	153 (36%)	109 (31%)	1.29	[0.95, 1.74]	0.0946
Nasal Discharge	378 (49%)	210 (50%)	168 (47%)	1.11	[0.84, 1.47]	0.4718
Facial Pain	161 (21%)	114 (27%)	47 (13%)	2.44	[1.67, 3.55]	0.6173
Cough	471 (61%)	307 (73%)	164 (46%)	3.15	[2.34, 4.25]	<0.001
Dyspnea	260 (33%)	182 (43%)	78 (22%)	2.71	[1.98, 3.73]	<0.001
Fever	339 (44%)	258 (61%)	81 (23%)	5.37	[3.92, 7.37]	<0.001
Diarrhea	299 (39%)	220 (53%)	79 (22%)	3.84	[2.80, 5.26]	0.0011

**Table 2 jcm-10-00570-t002:** Crude odds ratio (OR) for the associations between the VAS collected for each symptom and COVID-19 test. The VAS cut-off points that better predicted test positivity (by maximizing the Youden Index) are also shown, together with their sensitivity and specificity.

	OR	CI 95%		*p*-Value	VAS Cutoff Points	AUC	Sensitivity	Specificity
Loss of smell	10.85	7.47	15.77	<0.0001	21	0.76	0.60	0.88
Loss of taste	12.62	8.50	18.73	<0.0001	44	0.76	0.59	0.90
Nasal obstruction	1.75	1.19	2.56	0.0046	52	0.53	0.21	0.87
Nasal discharge	1.46	1.06	2.02	0.0232	41	0.52	0.30	0.77
Facial pain	2.54	1.73	3.74	<0.0001	15	0.57	0.26	0.88
Cough	3.19	2.36	4.31	<0.0001	3	0.65	0.73	0.54
Dyspnea	3.06	2.16	4.33	<0.0001	28	0.61	0.36	0.84

**Table 3 jcm-10-00570-t003:** Odds ratios (OR), 95% confidence intervals and *p*-values obtained for the step-wise multivariate logistic regression model under BIC criterion.

	OR	CI 95%		*p*-Value
Loss of smell	2.42	1.30	4.50	0.0053
Loss of taste	6.21	3.21	12.04	<0.001
Dyspnea	2.21	1.34	3.64	0.002
Fever	1.84	1.18	2.87	0.007
Diarrhea	2.02	1.29	3.16	0.002
Sex	3.11	1.97	4.90	<0.001

**Table 4 jcm-10-00570-t004:** Mean AUC, SE, SP, PPV, and NPV values obtained over the 50-fold cross-validation estimates for different datasets (Datasets D1–D5) and three different ML algorithms (LR, RF, and SVM).

		AUC	SE	SP	PPV	NPV
**Dataset 1 (11 predictors)** Loss of smell, loss of taste, nasal obstruction, nasal discharge, facial pain, cough, dyspnea, fever, diarrhea (Yes/No) Age and sex	**LR**	0.759	0.792	0.722	0.772	0.747
	**RF**	0.777	0.798	0.752	0.792	0.758
	**SVM**	0.771	0.811	0.724	0.777	0.764
**Dataset 2 (11 predictors)** Loss of smell, loss of taste, nasal obstruction, nasal discharge, facial pain, cough, dyspnea (VAS 0–100) Fever, Diarrhea (Yes/No) Age and sex	**LR**	0.777	0.783	0.771	0.803	0.751
	**RF**	0.798	0.818	0.775	0.812	0.782
	**SVM**	0.799	0.818	0.778	0.814	0.783
**Dataset 3 (11 predictors)** Loss of smell, loss of taste, nasal obstruction, nasal discharge, facial pain, cough, dyspnea (VAS > cutoff/VAS < cutoff) Fever, Diarrhea (Yes/No) Age and Sex	**LR**	0.773	0.787	0.757	0.794	0.751
	**RF**	0.791	0.812	0.765	0.804	0.775
	**SVM**	0.794	0.826	0.755	0.800	0.786
**Dataset 4 (4 predictors)** Loss of smell, loss of taste: VAS (0–100) Fever, Diarrhea (Yes/No)	**LR**	0.764	0.737	0.798	0.812	0.718
	**RF**	0.755	0.768	0.741	0.779	0.731
	**SVM**	0.759	0.784	0.730	0.776	0.742
**Dataset 5 (2 predictors)** Loss of Smell (VAS 0–100) Fever (Yes/No)	**LR**	0.768	0.812	0.715	0.772	0.762
	**RF**	0.768	0.797	0.733	0.780	0.753
	**SVM**	0.765	0.799	0.724	0.775	0.753

**Table 5 jcm-10-00570-t005:** Table summarizing the main characteristics of COVID-19 prediction models reported in the literature based on loss of smell and taste.

	Menni et al. [9]	Roland et al. [10]	Clemency et al. [14]	Kowall et al. [35]	Gerkin et al. [15]	Our Study
Sample	(UK)6452+/9186− (US)726+/2037−	145+/157−	225+/736−	296+/1641−	4148+/546−	421+/356−
Demographic Data	(UK) Positive group: Mean Age: 41.25 71.88% female Negative group: Mean Age: 43.2 76.40 % female	Mean age: 39 Sex: 72% female	N/A	Mean age: 53.5 years Sex: 61.3% females in the negative group and 57.8% in the positive group	Positive group: Mean Age: 40.6 74% female Negative group: Mean Age: 43.2 78% female	Positive group: Mean Age: 47.3 61% female Negative group: Mean Age: 45.2 78% female
Data collection	App-based symptom tracker	Public survey posted on social media	Nurse call center for healthcare workers (HCW)	Self-administered questionnaire	Online survey	Self-administered questionnaire
Variable Types	Categorical	Categorical	Categorical	Categorical	Categorical, continuous VAS	Categorical, Continuous VAS
Classification Methods	● Stepwise (forward and backward) ● Logistic Regression ● Akaike Information Criterion (AIC) Classifier threshold at 0.5	● Stepwise Logistic Regression ● (*p* = 0.05 for entry and 0.10 for removal with maximum iterations set at 20) Classifier threshold at 0.5	Logistic regression with maximum positive likelihood ratio (PLR) criterion	Stepwise backward logistic regression (*p* = 0.10 for entry and for removal)	L1 regularized logistic regression (penalty α = 1)	● Stepwise (forward and backward) ● Logistic Regression ● Bayesian Information Criterion (BIC) ● Random Forest (RF) ● Support Vector Machine (SVM) Classifier threshold at 0.5
Predictors	Age, sex, loss of smell and taste, severe or significant persistent cough, severe fatigue, skipped meals	(1) Smell or taste change, fever, body ache, shortness of breath, sore throat (2) Smell or taste change, fever and/or myalgia	(1) Fever, shortness of breath, dry cough (2) Fever, loss of taste or smell (3) Fever, shortness of breath, dry cough, loss of taste or smell	Age, sex, age, return from abroad, close contact with a confirmed case, the presence of fever, cough, exhaustion, taste or smell disorder, current smoking, general health condition and number of comorbidities	(1) Loss of smell, time duration (2) Model with 70 features	Five model datasets (see Table 4) including different variables among: age, sex, loss of smell, loss of taste, nasal obstruction, nasal discharge, facial pain, cough, dyspnea, fever and diarrhea
Validation method	● Holdout 80:20% ● training/test ● 10-fold cross-validation in the UK sample ● US validation sample	Holdout 75:25% training/test	N/A	Holdout 60:40% training-test	100-fold cross-validation with 80:20% training-test	● Holdout 75:25% training-test ● 50-fold cross-validation with ● 75:25% training-test
Accuracy Parameters	AUC = 0.76 SE = 0.66 SP = 0.83 PPV = 0.58 NPV = 0.87	(1) AUC = 0.82 SE = 0.56 (2) AUC = 0.75 SE = 0.70 SP = 0.73	(1) AUC = 0.63 SE = 0.93 SP = 0.09 (2) AUC = 0.75 SE = 0.89 SP = 0.48 (3) AUC = 0.77 SE = 0.98 SP = 0.08	AUC = 0.821	AUC = 0.72 SE = 0.85 SP = 0.75	AUC = 0.80 SE = 0.82 SP = 0.78 PPV = 0.81 NPV = 0.78

## Data Availability

The data presented in this study are available on request from the corresponding author. The data are not publicly available due to ethical reasons.

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
