# Peer review of "Loss of Smell and Taste Can Accurately Predict COVID-19 Infection: A Machine-Learning Approach"

_jcm, 2021, doi:10.3390/jcm10040570_

Round 1
Reviewer 1 Report
This study is based on 777 patients. Loss of smell and taste were found to be the symptoms with higher odds ratios of 6.21 and 2.42 for COVID-19 positivity. The ML algorithms applied reached an average accuracy of 80%, a sensitivity of 82%, and a specificity of 78% when using VAS to predict a COVID-19 diagnosis. This study concludes that smell and taste disorders are accurate predictors, with ML algorithms constituting helpful tools for COVID-19 diagnostic prediction.
I have some major and minor concerns:
Overall, olfactory pathway infection and concomitant smell dysfunction may be another etiology [ref. Lozada-Nur F. et al. 2020] of this infection. Based on our team’s on-going investigation into the pathophysiology of COVID-19 [PMID: 32818804, among others published work and a review article under review], we have seen that ocular manifestations [re. Wu P. et al., 2020] and SARS-CoV-2 presence in tears and conjunctival secretions [ref. Xia J. et al., 2019] have been reported. Eyes may provide an entry for the virus through the trigeminal nerve into the Central Nervus System. The involvement of facial, glossopharyngeal, and vagus nerves in the oral mucosa that is responsible for the transport of taste signals to the solitary nucleus in the brainstem may be the reason behind taste disturbance in up to 88% of the COVID-19 patients [ref. Lechien JR et al. 2020; ref #8 in the presented manuscript]. However, olfactory pathway infection and concomitant smell dysfunction may be another etiology [ref. Lozada-Nur F, et. Al. 2020]. The virus can invade the vagus nerve in respiratory and GI tracts [Li Z. 2020]. Given this current knowledge, this paper is timely and interesting and the fact that authors present novel data is a positive point.
However, the sample size is relatively small, especially with a pandemic where millions of patients have been infected. The concern is that with limited sample size the model performance can be higher than in reality. I understand that the authors tried using sampling as a way to address this limitation; however, this limitation has be said clearly in the manuscript.
The second major concern is the fact that if a patient with such signs and symptoms is misclassified and has actually COVID-19 if she/he was tested using PCR, then the misclassification would have caused this patient not be tested and potentially expose others. Therefore the cost for misclassification is asymmetric. Being classified to be tested for COVID and having a negative test results is not as serious, and for that reason, i wonder if the cut-off should be adjusted to optimize the model performance to reduce the change of such errors. Since the application of this tool is perhaps most appropriate for triaging patients at hospitals where resources are limited.
Minor edits:
1. Please do not use "lab" testing --> should always use the non-abbreviated form, that is “laboratory" testing
2. How missingness was handled is not clear. Please provide more details and clarification. If mean value was used to replace missing values for continuous variables and mode was used for categorical values, there is need for justification since these approaches tend to cause bias when applied to clinical data since missingness tend to be not completely at random, especially for variables with high missingness levels. It is not clear why citation 18 was used, which itself cites another article (general approach) for its imputation. Missingness levels for all the variables should be listed in the summary table. Also, missingness for cases and controls should be clarified as well. If a variable has higher missing in cases or in controls and the authors are using mean, then there is certainly a systemic bias in the dataset used for the ML.
3. Line 108-109: “by randomly splitting the sample on training (75%) and testing (25%) datasets.” --> I am not sure I understand this. Did the author perform multiple assessment by creating multiple training and testing sets? If so, more clarity is needed in the methodology. Either way, please clarify. From what I understand in line 117, “repeating the process by 50 each time” 50 different train/test combination were created.
4. 50 cross-validated estimates --> is this related to the 50 different train/test cohorts and the results from the evaluation based on the test dataset?
5. It would be useful to include correlation among various variables, especially since they are targeted in association with the olfactory pathway.
Some potentially useful citations that the authors are free to utilize as they see fit.
Lozada-Nur F, Chainani-Wu N, Fortuna G, Sroussi H. Dysgeusia in COVID-19: Possible Mechanisms and Implications. Oral Surg Oral Med Oral Pathol Oral Radiol [Internet]. 2020/06/27. 2020 Sep;130(3):344–6. Available from: https://pubmed.ncbi.nlm.nih.gov/32703719
Wu P, Duan F, Luo C, Liu Q, Qu X, Liang L, et al. Characteristics of Ocular Findings of Patients With Coronavirus Disease 2019 (COVID-19) in Hubei Province, China. JAMA Ophthalmol [Internet]. 2020 May 1;138(5):575–8. Available from: https://pubmed.ncbi.nlm.nih.gov/32232433
Xia J, Tong J, Liu M, Shen Y, Guo D. Evaluation of coronavirus in tears and conjunctival secretions of patients with SARS-CoV-2 infection. J Med Virol [Internet]. 2020/03/12. 2020 Jun;92(6):589–94. Available from: https://pubmed.ncbi.nlm.nih.gov/32100876
Lechien JR, Chiesa-Estomba CM, De Siati DR, Horoi M, Le Bon SD, Rodriguez A, et al. Olfactory and gustatory dysfunctions as a clinical presentation of mild-to-moderate forms of the coronavirus disease (COVID-19): a multicenter European study. Eur Arch Oto-Rhino-Laryngology [Internet]. 2020;277(8):2251–61. Available from: https://doi.org/10.1007/s00405-020-05965-1
Li Z, Liu T, Yang N, Han D, Mi X, Li Y, et al. Neurological manifestations of patients with COVID-19: potential routes of SARS-CoV-2 neuroinvasion from the periphery to the brain. Front Med [Internet]. 2020; Available from: https://doi.org/10.1007/s11684-020-0786-5
Author Response
Response to Reviewer 1 Comments
This study is based on 777 patients. Loss of smell and taste were found to be the symptoms with higher odds ratios of 6.21 and 2.42 for COVID-19 positivity. The ML algorithms applied reached an average accuracy of 80%, a sensitivity of 82%, and a specificity of 78% when using VAS to predict a COVID-19 diagnosis. This study concludes that smell and taste disorders are accurate predictors, with ML algorithms constituting helpful tools for COVID-19 diagnostic prediction.
I have some major and minor concerns:
Overall, olfactory pathway infection and concomitant smell dysfunction may be another etiology [ref. Lozada-Nur F. et al. 2020] of this infection. Based on our team’s on-going investigation into the pathophysiology of COVID-19 [PMID: 32818804, among others published work and a review article under review], we have seen that ocular manifestations [re. Wu P. et al., 2020] and SARS-CoV-2 presence in tears and conjunctival secretions [ref. Xia J. et al., 2019] have been reported. Eyes may provide an entry for the virus through the trigeminal nerve into the Central Nervus System. The involvement of facial, glossopharyngeal, and vagus nerves in the oral mucosa that is responsible for the transport of taste signals to the solitary nucleus in the brainstem may be the reason behind taste disturbance in up to 88% of the COVID-19 patients [ref. Lechien JR et al. 2020; ref #8 in the presented manuscript]. However, olfactory pathway infection and concomitant smell dysfunction may be another etiology [ref. Lozada-Nur F, et. Al. 2020]. The virus can invade the vagus nerve in respiratory and GI tracts [Li Z. 2020]. Given this current knowledge, this paper is timely and interesting and the fact that authors present novel data is a positive point.
Response 1: We thank the reviewer for his/her interest and comments about the novelty of our contribution to the knowledge of the etiology of the SARS-CoV2 infection. We also appreciate his/her contribution to improve the explanation on the etiology of the infection, according to their team´s ongoing investigation and the other references suggested. We have included all these considerations in the Discussion Section (Page 14, lines 363-375), and added the new references as [31-34].
Reviewer 2 Report
This is a well written case control study on 777 suspected Covid-19 patients.
The main issue regards the develop of a machine learning modelling framework in order to assess the predictive value of smell and taste disorders in Covid-19 disease.
The strenght of this paper is the implementing of various ML algorithms to test the importance of different predictor variables.
The use of VAS as evaluation tool, as well as the non-inclusion of asymptomatic patients tends to strongly influence all the predictive values.
Author Response
Response to Reviewer 2 Comments
This is a well written case control study on 777 suspected Covid-19 patients.
The main issue regards the develop of a machine learning modelling framework in order to assess the predictive value of smell and taste disorders in Covid-19 disease.
The strength of this paper is the implementing of various ML algorithms to test the importance of different predictor variables.
The use of VAS as evaluation tool, as well as the non-inclusion of asymptomatic patients tends to strongly influence all the predictive values.
Response: Thanks for these comments and the Reviewer’s interest in our manuscript. Regarding the last comment, we recognize that asymptomatic subjects from the general population were not systematically included in our study, due to the lack of PCR testing resources during the rapid spread of the pandemics in March in Spain. We have highlighted this as a major limitation of our study, probably increasing the predictive value of our models (Page 15, lines 427-432). Nevertheless, our criterion inclusion was that reported by the WHO for suspected cases, and therefore, our study did include 102 asymptomatic subjects (13% of the sample) who have had a contact with a confirmed positive case. This is also mentioned in (Page 5, lines 174).
Regarding the VAS, we have seen that these offer a higher mean discrimination accuracy (Fig. 3), which corroborates previous results in [15]. In addition, apart from VAS, we also tested different datasets including categorical predictors reporting only the presence or absence of symptoms, which still reached a high mean accuracy between 0.76 and 0.78 for different ML algorithms, thus corroborating the relevance of smell and taste disorders in COVID-19 prediction (Fig. 4). However, we also recognize in the Limitations Section that objective assessment of smell and taste cannot be replaced by VAS (Page 16, lines 442-444)
Reviewer 3 Report
The main question is effectiveness of using the sudden loss of smell / taste as a predictor for covid using a novel machine learning approach. It is a very interesting area as this draws on algorithms that can be used for mass use on a population based systems approach. The topic is not original in itself and many researchers have shown the prediction using the loss of smell and taste but the system approach used in original, in my opinion. It adds a fair amount to the knowledge on the subject and is well written. The area that I would suggest improvement is the explanation of the methodology which should make easy reading (sense) for the average clinician or reader who is not well versed with such algorithms.
This is a well written paper which confirms what researchers in the field have known for some time. Approaching it with the MLM is certainly unique and interesting. The one area where I would suggest improvement is the presentation of results and thereby drawing conclusions. For the average reader who is not well versed with the algorithms used for MLM it can be a confusing read which is a pity as there is so much good work being reported.
Author Response
Response to Reviewer 3 Comments
The main question is effectiveness of using the sudden loss of smell / taste as a predictor for covid using a novel machine learning approach. It is a very interesting area as this draws on algorithms that can be used for mass use on a population based systems approach. The topic is not original in itself and many researchers have shown the prediction using the loss of smell and taste but the system approach used in original, in my opinion. It adds a fair amount to the knowledge on the subject and is well written. The area that I would suggest improvement is the explanation of the methodology which should make easy reading (sense) for the average clinician or reader who is not well versed with such algorithms.
This is a well written paper which confirms what researchers in the field have known for some time. Approaching it with the MLM is certainly unique and interesting. The one area where I would suggest improvement is the presentation of results and thereby drawing conclusions. For the average reader who is not well versed with the algorithms used for MLM it can be a confusing read which is a pity as there is so much good work being reported.
Response: Thanks to the Reviewer for his/her comments and interest on our manuscript. We agree that the relevance of smell and taste disorders is a topic that has already been disclosed in the literature. As the reviewer highlights, the innovation of this study relies on confirming and furthering this previous evidence with a new comprehensive ML framework approach. Due to this main objective, we have focused on exhaustively describing the statistical and ML methods applied, which we acknowledge it can be a difficult read for the average clinician and/or reader not versed in the ML methodology.
Following the reviewer recommendation, we have re-written the method section of the manuscript trying to clarify the approach, page 4-5, lines 126-162. We have added a new figure to the manuscript (fig 1) with a flow chart of the methodology.
Round 2
Reviewer 1 Report
My concerns have been addressed.
Reviewer 2 Report
No other comments